# Variational- and Metric-based Deep Latent Space
# for Out-of-Distribution Detection

**Or Dinari**[1]

**Oren Freifeld**[1]

[1]The Department of Computer Science, Ben-Gurion University of the Negev, Be'er Sheva, Israel

## Abstract

One popular deep-learning approach for the task of Out-Of-Distribution (OOD) detection is based on thresholding the values of per-class Gaussian likelihood of deep features. However, two issues arise with that approach: first, the distributions are often far from being Gaussian; second, many OOD data points fall within the effective support of the known classes' Gaussians. Thus, either way it is hard to find a good threshold. In contrast, our proposed solution for OOD detection is based on a new latent space where: 1) each known class is well captured by a nearly-isotropic Gaussian; 2) those Gaussians are far from each other and from the origin of the space (together, these properties effectively leave the area around the origin free for OOD data). Concretely, given a (possibly-trained) backbone deep net of choice, we use it to train a conditional variational model via a Kullback Leibler loss, a triplet loss, and a new distancing loss that pushes classes away from each other. During inference, the class-dependent log-likelihood values of a deep feature ensemble of the test point are also weighted based on reconstruction errors, improving further the decision rule. Experiments on popular benchmarks show that our method yields state-of-the-art results, a feat achieved despite the fact that, unlike some competitors, we make no use of OOD data for training or hyperparameter tuning. Our code is available at https://github.com/BGU-CS-VIL/vmdls.

## 1 INTRODUCTION

Out-of-Distribution (OOD) detection is the following classification task. During training, there are labeled data points where each label is associated with one out of $C$ classes.

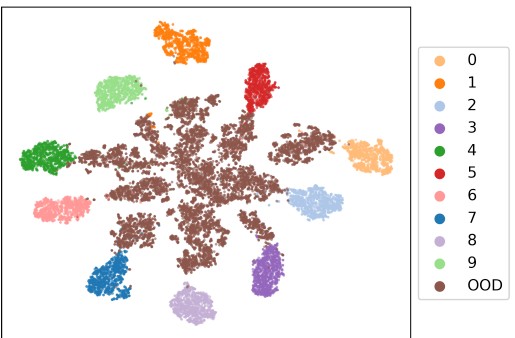

(a) Known classes: MNIST. OOD: Omniglot.

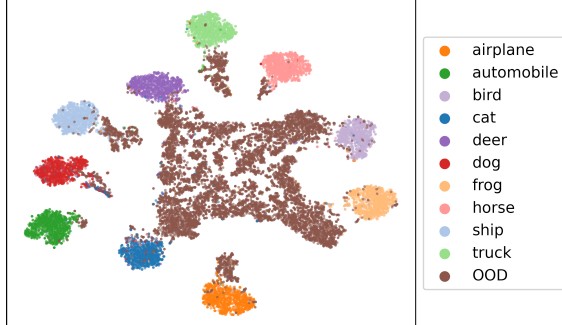

(b) Known classes: CIFAR-10. OOD: ImageNet-resize.

Figure 1: The t-SNE visualizations of feature ensembles, obtained using the proposed method, of test data in two different experiments (see text for details). The known classes are well separated from each other and from the OOD data.

The latter are referred to as *known classes* while the points from all the $C$ classes are collectively called *in-distribution train data*. In test time, in addition to previously-unseen (and unlabeled) points from the $C$ known classes – such points are called *in-distribution test data* – there are also (unlabeled) points that belong to neither of the known classes – such points are called *OOD data*. OOD detection is then the binary classification between in-distribution (test) data and OOD data. Note that in general, unlike the case of in-

*Accepted for the 38th Conference on Uncertainty in Artificial Intelligence* (UAI 2022).

distribution data, the availability of OOD data during the training cannot be assumed.

In Deep Learning (DL) [LeCun et al., 2015], OOD detection is especially important: *e.g.*, when Deep Neural Nets (DNNs) misclassify data, *they often do this with high confidence* [Goodfellow et al., 2015, Nguyen et al., 2015, Hendrycks and Gimpel, 2016, Moosavi-Dezfooli et al., 2017]. As such mistakes often stem from OOD data, it is beneficial to detect the latter.

Attempts to solve classification tasks, OOD detection included, are often based on trying to learn a low-dimensional latent space wherein classes are well separated. Many existing OOD-detection methods try to do so using metric learning (typically based on a contrastive loss) or a variational loss. Then, in test time, they use a decision rule based on thresholding the values of per-class Gaussian likelihoods. However, the problem with that Gaussian-based approach, at least when used naively, is twofold: 1) the distribution of each known class is usually not well captured by a Gaussian; 2), many of the OOD points tend to fall within the effective support of one or more of the known classes' Gaussians. Consequently, finding a good threshold is difficult.

Our proposed method eliminates these issues (see Figure 1), a feat achieved partly by leveraging in a synergistic way some of aforementioned ideas and partly by introducing new ideas.

For example, part of our method is based on using both a variational term and a metric loss. The variational term is a class-conditioned Gaussian Kullback-Leibler (KL) divergence loss [Sun et al., 2020a]. However, rather than using a contrastive loss, our metric loss consists of two terms: a triplet loss [Schroff et al., 2015] and a new loss called *distancing*. The distancing loss helps achieving a much better inter-class separation than what the triplet loss can do by itself. The key insight in this context is that without the distancing loss, the combination of the variational and triplet losses, which may work well in other classification tasks, tends to be less effective here (in OOD detection) as each loss pushes in a different direction, leading to insufficient separation and much of the OOD data ending up within the known classes' support. Adding the distancing term solves this, implicitly helping to form, between the known classes, an empty area wherein most of the OOD data ends up.

Another key difference between the learning stage of our method and others is that we measure our metric loss on the output of a *stochastic generator*, and not, as is usually done with such losses, on deterministic representations.

Our overall loss results in a new latent space, coined a *Variational- and Metric-based Deep Latent Space (VMDLS)*, wherein: 1) the distribution of each known class is (a nearly-isotropic) Gaussian; 2) these Gaussians are well separated from each other; 3) the in-distribution (train and test) points

are well separated from the OOD points.

After the DNN training and before the inference, we partially follow [Lee et al., 2018b] in that we extract a deep feature ensemble from our trained model for each of the training points, and fit, using a new implicit scheme, a (usually-anisotropic) Gaussian to the feature ensembles of each class. During inference, we compute for the feature ensemble of each test point its negative log-likelihood (ll) according to each of those Gaussians. Then, our method has two versions. In the basic one, denoted by **VMDLS$_b$**, we compare the best negative ll value against an application-specific threshold. As we will show, **VMDLS$_b$** is already a good OOD method – it has State-of-the-art (SOTA) results or close to it in terms of the F1 score (which takes in account both the classifier accuracy and OOD detection) – and achieves fairly-decent results in binary OOD detection. However, a better version, denoted by **VMDLS**, is obtained with the help of a reconstruction-based weighting scheme (see § 3.3). **VMDLS** yields results that are uniformly better than **VMDLS$_b$** and sets new SOTA results on several popular benchmarks in OOD detection (at least among methods that do not need OOD data for training/tuning). In Figure 1 we show, using t-SNE visualizations [Van der Maaten and Hinton, 2008], example results of the latent space associated with the learned feature ensemble in two different experiments: in one, the in-distribution data is MNIST [LeCun, 1998] and the OOD data is Omniglot [Lake et al., 2015]; in the second, the in-distribution data is CIFAR-10 [Krizhevsky, 2009] and the OOD data is ImageNet-resize [Liang et al., 2018]. All of the data shown in Figure 1 is test data.

**To summarize, our main contributions are:** 1) A new learning scheme of a novel latent space, where classes are well separated from each other and from OOD data based on a stochastic generator as well as a synergy between KL and metric losses via the help of a new metric-loss term; 2) A new inference scheme using that latent space, together with a reconstruction of low-level features, for OOD detection.

Together, these contributions give rise to a new end-to-end solution for OOD detection that yields SOTA results in multiple benchmarks and, importantly and unlike some existing methods, *requires no OOD data for training/tuning*.

Finally, all our experiments are reproducible, as the reader can verify by running our publicly-available code

## 2 RELATED WORK

Hendrycks and Gimpel [2016] proposed a popular OOD-detection baseline that uses a threshold on the maximal SoftMax score of a classifier to determine whether a point is OOD or not. ODIN [Liang et al., 2018] is a method that is applied to a previously-trained DNN and, by using small perturbations on the input and temperature scaling, calibrates the classifier's SoftMax scores, improving the separation

between in-distribution and OOD data. Lee et al. [2018b] proposed a method, called Mahalanobis, that extracts a feature ensemble from a trained DNN for all the known classes, and then fits class-conditioned Gaussian distributions (an idea we use too). Given a test point, they compute the Mahalanobis distance according to each of the Gaussians. Next, the decision (OOD or not) is made based on that distance, with the aid of a logistic-regression detector that they tune on a small validation set, which, in addition to in-distribution points, also *includes some OOD data*. Masana et al. [2018] proposed using metric learning for OOD detection. Specifically, they use a contrastive loss [Chen et al., 2020b] with OOD mining, incorporating an OOD set (different from the set they use at testing) during training. In [Winkens et al., 2020], a contrastive loss is used in several self-supervision tasks, together with a SoftMax-based classifier, for learning meaningful latent representations. In CSI [Tack et al., 2020] a contrastive loss is used as well, with shifting instances. Two other smart methods that use OOD data during training are Outlier Exposure (OE) [Hendrycks et al., 2019] and the energy-based model in [Liu et al., 2020].

More generally, note that the dependency of the elegant methods in [Hadsell et al., 2006, Hendrycks et al., 2019, Lee et al., 2018b, Liang et al., 2018, Liu et al., 2020] on OOD data for either tuning or training is a limitation. Moreover, that dependency also makes a comparison between these methods and ones that do not rely on OOD data (like ours) unfair against the latter.

Hsu et al. [2020] proposed Generalized ODIN, an ODIN variant that requires no hyperparameter tuning on OOD data. Recently, Zaeemzadeh et al. [2021] proposed using 1-dimensional subspaces for OOD detection; similarly to Generalized ODIN, their method requires no tuning on OOD data. Yoshihashi et al. [2019] proposed CROSR, where instead of using only the DNN's output, they also used a latent representation extracted from the DNN. They have also introduced DHRNet, a DNN which assists the learning of meaningful representations via reconstruction.

While methods that rely on trained classifiers (*e.g.*, [Liang et al., 2018, Lee et al., 2018b, Hendrycks and Gimpel, 2016]) have proven to be successful, they are, as was noted in [Lee et al., 2018a], limited by the trained classifier. In contrast, rather than relying on such trained DNNs, our method is based on a novel training scheme. Our method may be viewed as belonging to the hybrid discriminative-generative camp, an approach adopted by several recent works such as OpenHybrid [Zhang et al., 2020], OSAD [Shao et al., 2020] (which is used for adversarial defense) and SSD [Sehwag et al., 2020] (which utilizes unlabeled in-distribution samples).

A reconstruction-based method was proposed by Perera et al. [2020] who trained a generative model for the known classes, and augmented the input with representations obtained from

that model for training a classifier. They also used self-supervision to learn more informative features. Chen et al. [2020a] used Reciprocal Point Learning (RPL) to learn compact and discriminative representations. More relevant to our work is a method called CGDL [Sun et al., 2020a] which trains a Variational Auto Encoder (VAE) with a class-conditioned Gaussian distribution. Particularly, their model learns a different Gaussian for each class and uses a ladder architecture to extract high-level features, a SoftMax-based classifier, and a detector of unknown classes in the learned latent space. From their work we have adopted (and adapted) a single idea and that is the usage of class-conditioned Gaussians. Another recent work employing a conditional VAE is CVAECapOSR [Guo et al., 2021] which is based on fitting to each class a predefined Gaussian. One key difference between our usage of a Conditional VAE and [Sun et al., 2020a, Guo et al., 2021] is that while they use predefined target Gaussians [Guo et al., 2021], or explicitly learn the target Gaussians [Sun et al., 2020a], we learn them implicitly by the metric losses. This helps us achieve better separation between the classes.

# 3 THE PROPOSED METHOD

We design our model to have a deep latent space with two desiderata: 1) That the empirical distribution of each class will be well approximated by a Gaussian (this simplifies inference and implies that the level sets of each class-dependent empirical distribution are nearly convex). 2) That each such Gaussian will be far from the Gaussians of all of the other classes. Once these properties are achieved, a likelihood-based decision whether a point belongs to a known class or not is easily made, as we explain below.

Let $D$ be the dimension of the data, let $\boldsymbol{x} \in \mathbb{R}^D$ denote a data point, and let $d < D$ be the dimension of the sought-after latent space. Our model is an encoder (an $\mathbb{R}^D \to \mathbb{R}^d$ function) that consists of a backbone DNN denoted by $\boldsymbol{g}$ : $\mathbb{R}^D \to \mathbb{R}^k$ (where $d < k < D$) and a stochastic generator, $\boldsymbol{f} : \mathbb{R}^k \to \mathbb{R}^d$, that generates $d$-dimensional samples and whose functionality is closely related to the sampling step in a VAE [Kingma and Welling, 2014]. More concretely, let

$$\boldsymbol{f}(\boldsymbol{g}(\boldsymbol{x})) = \boldsymbol{\mu}(\boldsymbol{g}(\boldsymbol{x})) + (\boldsymbol{\sigma}(\boldsymbol{g}(\boldsymbol{x})) \odot \boldsymbol{\epsilon}) \in \mathbb{R}^d$$
$$\boldsymbol{\mu} = (\mu_1, \ldots, \mu_d) : \mathbb{R}^k \to \mathbb{R}^d$$
$$\boldsymbol{\sigma} = (\sigma_1, \ldots, \sigma_d) : \mathbb{R}^k \to \mathbb{R}^d_{>0}$$
$$\boldsymbol{\epsilon} \sim \mathcal{N}(\boldsymbol{0}_{d \times 1}, \boldsymbol{I}_{d \times d}) \tag{1}$$

where $\odot$ is the Hadamard (element-wise) product. Each of the functions $\boldsymbol{\mu}$ and $\boldsymbol{\sigma}$ is defined via its own fully-connected layer whose input is $\boldsymbol{g}(\boldsymbol{x})$ (the positivity of $\boldsymbol{\sigma}$ is ensured by exponentiating the output of its layer). A difference from a standard VAE is that here $\boldsymbol{f}(\boldsymbol{g}(\boldsymbol{x})) \in \mathbb{R}^d$ is never transformed back to $\mathbb{R}^D$; *i.e.*, there is no "decoding" as we do not try to reconstruct $\boldsymbol{x}$. The role of the latent

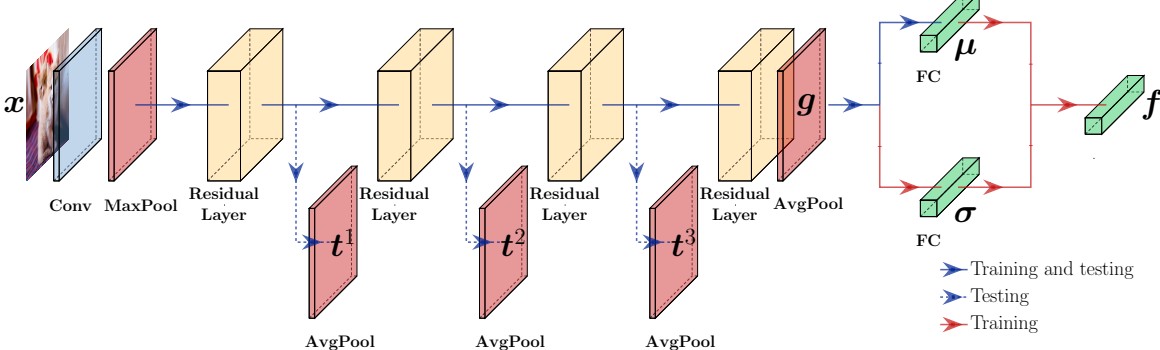

Figure 2: The proposed model with a ResNet backbone.

space will become apparent when we discuss our loss function. As for $\boldsymbol{g}$, in principle any backbone could be used within our method. In our experiments we explored several types: ResNet18 [He et al., 2016]; ResNet34 [He et al., 2016]; DenseNet-BC100 [Huang et al., 2017]; WideResNet28 [Zagoruyko and Komodakis, 2016]; VGG [Simonyan and Zisserman, 2015]; a simple Convolutional Neural Net (CNN). The values of $k$ and $d$ depend on the backbone and dataset; for the concrete values used in our experiments see our Supplemental Material (**SupMat**). The model's architecture (with a ResNet backbone) is shown in Figure 2, where (as we explain in the next sections) the red lines are used only during training, the solid blue lines are used during both training and testing, and the dashed blue lines are used only during testing.

## 3.1 TRAINING

Recall that $C$ is the number of known classes. Let $N$ be the number of training points and let $y_i \in \{1, \ldots, C\}$ be the class label of a training point $\boldsymbol{x}_i \in \mathbb{R}^D$. We split the training data into batches using the following standard mining strategy. For each batch we choose $B_C$ classes at random, where $B_C \in \{2, \ldots, C\}$ (*i.e.*, there must be representatives from at least two classes in each batch), and then, from each such a randomly-chosen class we choose $B_N$ examples at random. These examples constitute the batch. The values of $B_C$ and $B_N$ depend on both the backbone and the dataset (the concrete values used in our experiments appear in the **SupMat**). Our overall loss combines three loss types. The first loss is the popular triplet loss [Schroff et al., 2015] which is based on three input examples (from the same batch) at each time: an anchor example, $\boldsymbol{x}_i^a$; a positive example, $\boldsymbol{x}_i^p$; a negative example, $\boldsymbol{x}_i^n$. This loss aims to make (in the latent space) the squared $\ell_2$ distance between $\boldsymbol{x}_i^a$ and $\boldsymbol{x}_i^p$ smaller than the squared $\ell_2$ distance between $\boldsymbol{x}_i^a$ and $\boldsymbol{x}_i^n$ by, at least, a user-defined margin $M_t$:

$$\mathcal{L}_t(\boldsymbol{x}_i^a, \boldsymbol{x}_i^p, \boldsymbol{x}_i^n) \triangleq \max(0, M_t + \|\boldsymbol{f}(\boldsymbol{g}(\boldsymbol{x}_i^a)) - \boldsymbol{f}(\boldsymbol{g}(\boldsymbol{x}_i^p))\|^2 - \|\boldsymbol{f}(\boldsymbol{g}(\boldsymbol{x}_i^a)) - \boldsymbol{f}(\boldsymbol{g}(\boldsymbol{x}_i^n))\|^2). \quad (2)$$

The anchor and the positive examples are chosen from the same class, while the negative example is from some other class. The rationale is to push examples of different classes away from each other while drawing same-class examples closer to each other. While this is desired in our context as well, it turns out that it is not enough for pushing the classes *sufficiently far* from each other. In other words, we want to push the anchor away from the negative examples independently of the current distance between $\boldsymbol{x}_i^a$ and $\boldsymbol{x}_i^p$. We thus propose adding a new simple metric loss term, called *distancing*. It uses the same $\boldsymbol{x}_i^a$ and $\boldsymbol{x}_i^n$ from the nominal triplet used in $\mathcal{L}_t$ (and ignores $\boldsymbol{x}_i^p$) and has its own user-defined margin parameter, $M_d$:

$$\mathcal{L}_d(\boldsymbol{x}_i^a, \boldsymbol{x}_i^n) \triangleq \max(0, M_d - \|\boldsymbol{f}(\boldsymbol{g}(\boldsymbol{x}_i^a)) - \boldsymbol{f}(\boldsymbol{g}(\boldsymbol{x}_i^n))\|^2). \quad (3)$$

The idea here is to further push the anchor from the negative examples, without affecting its distances from the positive ones. As an aside, the addition of the $M_d$ hyperparameter does not increase the number of knobs to tweak; as it turns out, the presence of the $\mathcal{L}_d$ term lets us fix the value of $M_t$ (the margin from the standard triplet loss) to $M_t = 0.1$ in all our experiments on all the datasets (changing the value of $M_t$ is equivalent to scaling all the other hyperparameters accordingly). As for the value of $M_d$, we usually use $M_d = d$. Lastly, our combined *metric loss* for triplet $(\boldsymbol{x}_i^a, \boldsymbol{x}_i^p, \boldsymbol{x}_i^n)$, *measured on the output of the stochastic generator, $\boldsymbol{f}$*, is:

$$\mathcal{L}_m(\boldsymbol{x}_i^a, \boldsymbol{x}_i^p, \boldsymbol{x}_i^n) = \mathcal{L}_t(\boldsymbol{x}_i^a, \boldsymbol{x}_i^p, \boldsymbol{x}_i^n) + \mathcal{L}_d(\boldsymbol{x}_i^a, \boldsymbol{x}_i^n). \quad (4)$$

The third loss term is a Gaussian class-conditioned KL divergence loss, proposed by [Sun et al., 2020a] (based on [Kingma and Welling, 2014]). Its purpose is to make the intra-class distribution in the latent space as close as possible to an isotropic Gaussian. The KL-divergence loss (*i.e.*, the negated KL divergence) for $\boldsymbol{x}_i$ is

$$\mathcal{L}_{\mathrm{KL}}(\boldsymbol{x}_i) = -D_{\mathrm{KL}}(q(\cdot|\boldsymbol{g}(\boldsymbol{x}_i))\|p(\cdot|y_i)) \quad (5)$$

where $q(\cdot|\boldsymbol{g}(\boldsymbol{x}_i))$ is a $d$-dimensional Gaussian probability density function (pdf) with a mean vector $\boldsymbol{\mu}(\boldsymbol{g}(\boldsymbol{x}_i))$ and a diagonal covariance matrix whose $(j, j)$ entry is

$\sigma_j^2(\boldsymbol{g}(\boldsymbol{x}_i))$ while $p(\cdot|y_i)$ is an isotropic $d$-dimensional Gaussian pdf, associated with class $y_i$, with a mean vector $\boldsymbol{m}(y_i) = (m_1(y_i), \ldots, m_d(y_i))$ and variance $s^2$ (we use $s^2 = 0.1$ in all classes and in all our experiments; changing $s^2$ to another constant will only change the scaling of the other hyper-parameters). We now explain how $(\boldsymbol{m}(c))_{c=1}^C$ are defined. While in [Guo et al., 2021] the means are predefined, and in [Sun et al., 2020a] the means were learned at the same time with the rest of the model, we compute them based on the previous epoch. That is, at epoch $t$, for each class $c$ we set $\boldsymbol{m}(c) = \frac{1}{N_c} \sum_{\boldsymbol{x}_i : y_i = c} \boldsymbol{f}_{t-1}(\boldsymbol{g}(\boldsymbol{x}_i))$, where $N_c = |\{i : y_i = c\}|$ is the number of training points in class $c$, and $\boldsymbol{f}_{t-1}(\boldsymbol{g}(\boldsymbol{x}_i))$ is the previous epoch's output for $\boldsymbol{x}_i$.

The reason for using the fixed results from the previous epoch is that since the *distancing loss* pushes the classes away from each other, learning the target means from the previous epoch gradually pushes the target Gaussians from each other, and does so without the added complexity of learning them separately (as was done in [Sun et al., 2020a]). This encourages the classes to have Gaussian distributions of similar shape and (small) size in the latent space, simplifying learning and inference. Based on the above, it can be shown (**SupMat**) that $\mathcal{L}_{\mathrm{KL}}(\boldsymbol{x}_i)$ equals

$$\sum_{j=1}^d -\log \frac{s}{\sigma_j(\boldsymbol{g}(\boldsymbol{x}_i))} - \frac{\sigma_j^2(\boldsymbol{g}(\boldsymbol{x}_i)) + (\mu_j(\boldsymbol{g}(\boldsymbol{x}_i)) - m_j(y_i))^2}{2s^2} + \frac{1}{2}. \quad (6)$$

Lastly, $\mathcal{L}(\boldsymbol{x}_i)$, the overall loss of $\boldsymbol{x}_i$, and $\mathcal{L}((\boldsymbol{x}_i)_{i=1}^N)$, the loss across the entire training data, are

$$\mathcal{L}(\boldsymbol{x}_i) = \mathcal{L}_{KL}(\boldsymbol{x}_i) + \frac{1}{|\mathcal{T}(\boldsymbol{x}_i)|} \sum_{\boldsymbol{x}_i^p, \boldsymbol{x}_i^n \in \mathcal{T}(\boldsymbol{x}_i)} \mathcal{L}_m(\boldsymbol{x}_i, \boldsymbol{x}_i^p, \boldsymbol{x}_i^n)$$

$$\text{and } \mathcal{L}((\boldsymbol{x}_i)_{i=1}^N) = \frac{1}{N} \sum_{i=1}^N \mathcal{L}(\boldsymbol{x}_i) \quad (7)$$

where $\mathcal{T}(\boldsymbol{x}_i)$ consists of all of the possible triplets for $\boldsymbol{x}_i$ in the batch.

At this point, the reader may wonder about the role of the stochastic generator as seemingly the metric loss can be measured on the deterministic $\boldsymbol{\mu}(\boldsymbol{g}(\cdot))$ instead of the stochastic $\boldsymbol{f}(\boldsymbol{g}(\cdot))$. However, empirically, the deterministic option rarely works well (and even then it is no better than the stochastic one) while in most cases it led to unstable optimization and poor results. We did not encounter these issues with the stochastic version. A plausible theoretical explanation (similar to one used in the classical VAE) to this phenomenon is that the sampling leads, during the back propagation, to unbiased estimates of the true gradient.

Importantly, the effect of the overall loss is not only creating small isotropic Gaussians that are far from each other but also creating a large empty area between them (for an intuitive explanation, see **SupMat**). Empirically, this is where the OOD data ends up during test time.

**Learning $C$ Anisotropic Gaussians over Deep Feature Ensembles.** After the training, following an idea from [Lee

et al., 2018b], we run a forward pass on the training set (with no augmentations) and for each $\boldsymbol{x}_i$ collect a feature ensemble, $\boldsymbol{t}_i$:

$$\boldsymbol{t}_i \triangleq \boldsymbol{t}(\boldsymbol{x}_i) = (\boldsymbol{t}^1(\boldsymbol{x}_i), \boldsymbol{t}^2(\boldsymbol{x}_i), \boldsymbol{t}^3(\boldsymbol{x}_i), \boldsymbol{g}(\boldsymbol{x}_i), \boldsymbol{\mu}(\boldsymbol{g}(\boldsymbol{x}_i)) \quad (8)$$

where $(\boldsymbol{t}^1, \boldsymbol{t}^2, \boldsymbol{t}^3)$ are the outputs (after performing average pooling) of 3 different layers from the backbone. The choice of those layers depends on the backbone's architecture. In block-based architectures (*e.g.*, ResNet or DenseNet) we use the outputs of the first 3 blocks; see, *e.g.*, Figure 2. In other architectures, various choices may be made. See **SupMat** for examples in our experiments. Next, for each class $c$, we fit a multivariate *anisotropic* Gaussian to $(\boldsymbol{t}_i)_{i:y_i=c}$, the feature ensembles associated with that class.

## 3.2 TESTING

Given a test point $\boldsymbol{x} \in \mathbb{R}^D$ we extract its feature ensemble $\boldsymbol{t}$ and calculate its log-likelihood values according to each of the $C$ Gaussians from § 3. Let $(l_c)_{c=1}^C$ denote these values and let $l = \max_c l_c$ and $c_{\max} = \arg\max_c l_c$. The t-SNE visualizations (Figure 1) of the feature ensembles for the *test data* in two benchmarks show that the known classes are well separated from each other as well as from the OOD data; *i.e.*, the properties of the last latent layer propagate to the other latent layers, hence also to the additional features in the ensemble. Moreover, likelihood values of the OOD points are in general much lower than those of the in-distribution test points. In principle we can stop here, and use the following rule to decide whether $\boldsymbol{x}$ belongs to class $c_{\max}$ or if it is an OOD point: $\widehat{y} = c_{\max}$ if $l > \lambda$ and $\widehat{y} = \text{OOD}$ otherwise where $\widehat{y}$ is the predicted label of $\boldsymbol{x}$. Here, $\lambda$ is a user-defined threshold whose application-specific value can be determined based on either the train set or a small validation set (neither of these sets contains OOD data) according to the allowed False Negative Rate (FNR) while taking into account the model accuracy (if interested in classification as well as OOD detection). That is, $\lambda$ is chosen according to the relative importance the user gives to False Positives (FP) or False Negatives (FN). We refer to this basic version of our method as **VMDLS$_b$**. As we show in § 4, **VMDLS$_b$** already achieves SOTA results when benchmarking F1 score (taking in account both classification and OOD) or close to it, and has good results in binary OOD detection. That said, a simple change improves results even further; see § 3.3.

## 3.3 RECONSTRUCTION-BASED WEIGHTING

To improve results we use and adapt a popular technique [Sun et al., 2020a,b, Yoshihashi et al., 2019] based on test-point reconstruction. Concretely, we train, on the same training data (but independently of our model) a simple (non-conditional, non-variational) autoencoder (see **SupMat** for details) with an $\ell_2$ reconstruction loss. Let $\widehat{\boldsymbol{x}}$ denote

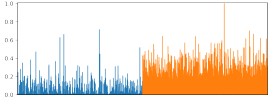 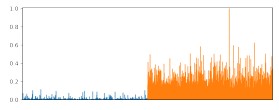

(a) Unmodified values
(**VMDLS**$_b$)

(b) Modified values (**VMDLS**)

Figure 3: Negative ll (scaled to one) without (a) and with (b) the modification. In-distribution: CIFAR-10 (blue). OOD: ImageNet-resize (orange). Note the better separation in (b).

the reconstruction of test point $x$ using that AE. We run a forward pass of our model on both $x$ and $\widehat{x}$. Let $t^1$ and $\widehat{t^1}$ denote the first features (see Eq. (8)) associated with $x$ and $\widehat{x}$, respectively, and let $w = ||t^1 - \widehat{t^1}||_{\ell_2}^2$. If $x$ is in-distribution, then $w$ is usually small. We now modify $l$ (from § 3.2) by $w$: *i.e.*, $l_{\text{new}} \triangleq w \cdot l$. Within the modified values, the separation of OOD points from the rest is better than before, a fact becoming clearer when the negative log-likelihood values are normalized to lie in the unit interval and a similar procedure is applied to their modified version: while most of the OOD values are hardly affected by the modification, the in-distribution values are vastly reduced; see Figure 3. This leads to a new rule: $\widehat{y} = c_{\max}$ if $l_{\text{new}} > \lambda$ and $\widehat{y} = \text{OOD}$ otherwise. We refer to our method, when using this rule, as **VMDLS**.

One difference in how we use the reconstruction-based weighting from how such schemes are commonly used is that we rely of the reconstruction of the (relatively) shallow feature. The reason is that in the deeper features, the differences between the OOD and in-distribution are (empirically) less noticeable as many fine details get washed out. For some intuition, suppose that in-distribution points are cars and the OOD points are animals. When a dog image enters the simple AE (trained on cars) its reconstruction is poor; *e.g.*, it might look like a blurry/unrecognizable car. Both the dog image and its poor reconstruction are fed to our model. The low-level features (*e.g.*, edges) in both cases will still be fairly-well captured but will be very different across the two images of the dog and its poorly-reconstructed version. That disparity, however, becomes smaller when moving deeper into the model. This is because the (true) high-level features of the dog image (*e.g.*, the dog's legs) are poorly captured by our model (learned on cars) and will appear in neither the dog's image nor its reconstructed version. Thus, our reconstruction-based weighting is based on the shallow features. See **SupMat** for empirical comparison with deeper features.

## 4 EXPERIMENTS AND RESULTS

We evaluated our method on several OOD-detection benchmarks and compared it with several key relevant methods, focusing on methods whose authors made their code available and/or published results on those benchmarks. We excluded methods that use OOD data during training (*e.g.*, OE [Hendrycks et al., 2019]) as such methods enjoy an unfair advantage. We did include, however, two methods that use OOD data for hyperparameter tuning (ODIN [Liang et al., 2018] and Mahalanobis [Lee et al., 2018b]). In our experiments, we followed testing methodologies employed in recent works [Sun et al., 2020a,b, Yoshihashi et al., 2019, Oza and Patel, 2019, Liang et al., 2018, Lee et al., 2018b, Hsu et al., 2020]. We split the experiments to two types. In the first, the binary OOD detection benchmark, we used two widely-used metrics: 1) The Area Under the Receiver Operating Characteristic curve (AUROC), which is a threshold-independent metric (unlike, *e.g.*, the F1 score). 2) The true-negative rate at a true-positive rate of $0.95$, denoted as TNR@TPR95. For the second, we evaluated our method on the popular Open Set Recognition (OSR) [Scheirer et al., 2012] task, in which one measures how well a method predicts (in the test data) the $C + 1$ labels, where the $(C + 1)$-th label represents the OOD data. To that aim, and as is common, we computed the F1 score per class, and then computed the macro-average across those $C + 1$ values.

We experimented with several backbones: ResNet18; ResNet34; WideResNet28; DenseNet-BC100; a modified version of VGG (as defined in [Yoshihashi et al., 2019]); a plain CNN (as defined in [Yoshihashi et al., 2019]). For each model and dataset we used a suitable (and fairly-standard) training strategy (except in the CIFAR-100 DenseNet and the CIFAR-10 WideResnet28 experiments where we fine-tuned a network pretrained on the data with a standard Soft-Max classifier, in all the other experiments we trained the models from scratch); see **SupMat** for details.

**Benchmarks.** For **Binary OOD detection**, we used the benchmarks proposed in [Liang et al., 2018]: using CIFAR-10 [Krizhevsky, 2009] or CIFAR-100 [Krizhevsky, 2009] for the known classes, and either of the following 4 datasets as OOD: ImageNet-crop; ImageNet-resize; LSUN-crop; LSUN-resize. Thus, there were 8 experiments (one per combination). ImageNet-crop and ImageNet-resize are subsets of ImageNet [Deng et al., 2009], either cropped or resized to the appropriate size for CIFAR. Likewise, LSUN-crop and LSUN-resize are cropped/resized subsets of LSUN [Yu et al., 2015]. Each of the OOD sets contains $10K$ images, curated by [Liang et al., 2018]. For **OSR**, in one type of experiments we used CIFAR-10 as the known classes, and the same aforementioned 4 OOD sets (so there were 4 such experiments). We also did three experiments using MNIST [LeCun, 1998] as the known classes, and one of the following three OOD sets each time: Omniglot's test set [Lake et al., 2015], which contains different handwritten characters from multiple alphabets; MNIST-noise, a version of MNIST with random uniform noise added to it; NOISE, where each observation is a random uniform noise. Each OOD set contains $10K$ images.

Table 1: OOD-detection results on CIFAR-10 and CIFAR-100. Results: macro-averages over 4 OOD datasets: ImageNet-resize; ImageNet-crop; LSUN-crop; LSUN-resize. Backbones: ResNet18 for CSI; WideResnet28 for SubSpaces; DenseNet-BC100 for the rest.

| | CIFAR-10 | | CIFAR-100 | |
|---|---|---|---|---|
| Method | AUROC | TNR@TPR95 | AUROC | TNR@TPR95 |
| ODIN[1] | 0.987 | 0.945 | 0.910 | 0.599 |
| Mahalanobis[2] | **0.993** | 0.979 | 0.986 | 0.940 |
| DeConf-C | 0.989 | 0.946 | 0.820 | 0.556 |
| Mahalanobis* | 0.962 | 0.820 | 0.916 | 0.649 |
| ODIN* | 0.904 | 0.556 | 0.911 | 0.581 |
| CSI | 0.981 | 0.897 | 0.913 | 0.582 |
| SubSpaces | 0.988 | 0.943 | 0.930 | 0.675 |
| VMDLS$_b$ (Ours) | 0.977 | 0.866 | 0.975 | 0.878 |
| VMDLS (Ours) | 0.990 | **0.982** | **0.995** | **0.980** |

[1] Uses an OOD set (though not the OOD test set) for tuning.
[2] Uses $1/10$ of the test and OOD sets for tuning and training a linear regressor during inference.

Table 2: Macro-average F1-scores of 11 classes (10 known; 1 unknown) on CIFAR-10 as the known classes and 4 different OOD sets.

| Method | Backbone | ImageNet-c | ImageNet-r | LSUN-c | LSUN-r |
|---|---|---|---|---|---|
| SoftMax | VGG | 0.639 | 0.653 | 0.642 | 0.647 |
| OpenMax | VGG | 0.660 | 0.684 | 0.657 | 0.668 |
| CROSR | VGG | 0.721 | 0.735 | 0.720 | 0.749 |
| C2AE | CNN | 0.837 | 0.826 | 0.783 | 0.801 |
| CGDL | VGG | 0.840 | 0.832 | 0.806 | 0.812 |
| CPGM-AAE | VGG | 0.793 | 0.830 | 0.820 | 0.865 |
| RPL | WResNet | 0.811 | 0.810 | 0.846 | 0.820 |
| CVAECapOSR | ResNet34 | 0.857 | 0.834 | 0.868 | 0.882 |
| ODIN[1] | DenseNet | 0.910 | 0.904 | 0.898 | 0.911 |
| VMDLS$_b$ (Ours) | VGG | 0.850 | 0.845 | 0.849 | 0.844 |
| VMDLS$_b$ (Ours) | ResNet18 | 0.889 | 0.875 | 0.888 | 0.882 |
| VMDLS$_b$ (Ours) | DenseNet | 0.913 | 0.908 | 0.914 | 0.911 |
| VMDLS$_b$ (Ours) | WResNet | 0.934 | 0.901 | 0.935 | 0.910 |
| VMDLS (Ours) | VGG | 0.907 | 0.861 | 0.900 | 0.886 |
| VMDLS (Ours) | ResNet18 | 0.926 | 0.906 | 0.904 | 0.924 |
| VMDLS (Ours) | ResNet34 | 0.930 | 0.911 | 0.928 | 0.907 |
| VMDLS (Ours) | DenseNet | 0.939 | 0.927 | 0.941 | 0.934 |
| VMDLS (Ours) | WResNet | **0.945** | **0.930** | **0.942** | **0.936** |

[1] Uses an OOD set (though not the OOD test set) for tuning.

**Binary OOD Detection.** On the binary OOD-detection benchmark described above, we compared our method against 5 other methods: ODIN [Liang et al., 2018]; Mahalanobis [Lee et al., 2018b]; DeConf-C [Hsu et al., 2020];CSI [Tack et al., 2020]; SubSpaces [Zaeemzadeh et al., 2021]. Note this comparison is biased in favor of ODIN and Mahalanobis; unlike the other methods (ours included) which require no tuning on OOD data, ODIN requires tuning on a different OOD set, while Mahalanobis uses $\frac{1}{10}$ of both the test sets of the known classes and the OOD to tune its hyperparameters. To account for that, we also include the results for versions of those methods which do not depend on OOD sets for tuning, as described by Hsu et al. [2020]. We mark them as ODIN* and Mahalanobis*. We report the results in Table 1 (the numbers for ODIN*, Mahalanobis* and DeConf-C are taken from [Hsu et al., 2020]), where we show the macro-average results across the 4 OOD sets (full results are in the **SupMat**). It is observable that in most cases **VMDLS** outperforms the others. The exception is the AUROC score for CIFAR-10 where Mahalanobis has a slightly higher score than **VMDLS**; however, once its usage of OOD sets is denied (*i.e.*, Mahalanobis*), its results drop below ours.

**OSR.** On the **CIFAR-10-as-in-distribution benchmark** described above, we compared with the following methods: SoftMax; OpenMax [Bendale and Boult, 2016]; CROSR [Yoshihashi et al., 2019], C2AE [Oza and Patel, 2019]; ODIN [Liang et al., 2018]; CGDL [Sun et al., 2020a]; CPGM-AAE [Sun et al., 2020b] RPL [Chen et al., 2020a]; CVAECapOSR [Guo et al., 2021]. Unlike in the previous benchmark (where most methods used DenseNet-BC100), here some methods use different backbones and, in addition, ODIN uses a different OOD set for hyperpa-

rameter tuning, giving it an advantage. For a fair comparison, we have evaluated our method with several different backbones used by the other methods: 1) the same modified VGG used in [Yoshihashi et al., 2019, Sun et al., 2020a,b]; 2) ResNet34 [He et al., 2016] (used in [Guo et al., 2021]); 3) DenseNet-BC100 (used in [Liang et al., 2018]); 4) WideResNet28 (used in [Zaeemzadeh et al., 2021]); 5) For completeness, we also used the popular ResNet18 [He et al., 2016]. On the **MNIST-as-in-distribution benchmark** described above, we compared our method with 5 others: A standard SoftMax; OpenMax [Bendale and Boult, 2016]; CROSR [Yoshihashi et al., 2019]; CPGM-VAE [Sun et al., 2020a]; CPGM-AAE [Sun et al., 2020b]; CVAECapOSR [Guo et al., 2021]. For a fair comparison, as the backbone for our method we used the same CNN specified in [Yoshihashi et al., 2019]. Table 2 and Table 3 summarize the results. The results in Table 2 show that, for a given backbone, **VMDLS** outperforms the other methods that use that backbone. Moreover, even with ResNet18 **VMDLS** outperforms the DenseNet-based ODIN (despite ODIN's usage of tuning on OOD data). Likewise, the results in Table 3 show that **VMDLS** outperforms the others by a large margin. Finally, note that in Table 2, after **VMDLS**, **VMDLS$_b$** is almost always the runner-up.

**Remarks. 1)** As ODIN's published code (targeting binary OOD detection) was easy to adapt for OSR we included it in these experiments. **2)** F1-score is threshold-dependent. For the purpose of benchmarking, the F1-scores values in this paper were computed (as was done, *e.g.*, in [Sun et al., 2020a]) using a threshold that aims to recognize 95% of the train set as belonging to known classes. **3)** Except our own results and ODIN's results (which we computed our-

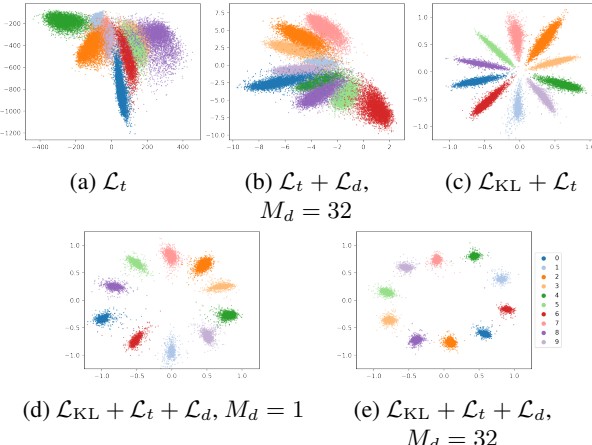

(a) $\mathcal{L}_t$     (b) $\mathcal{L}_t + \mathcal{L}_d$, $M_d = 32$     (c) $\mathcal{L}_{\mathrm{KL}} + \mathcal{L}_t$

(d) $\mathcal{L}_{\mathrm{KL}} + \mathcal{L}_t + \mathcal{L}_d$, $M_d = 1$     (e) $\mathcal{L}_{\mathrm{KL}} + \mathcal{L}_t + \mathcal{L}_d$, $M_d = 32$

Figure 4: The effect of the losses on $(\boldsymbol{\mu}(\boldsymbol{x}_i))_{i=1}^N$ in MNIST's train data with $d = 2$ (*i.e.*, this is *not* a t-SNE plot but the actual features). Each digit has its own color.

selves), the rest of the results in Table 2 and Table 3 were reported in [Sun et al., 2020b,a, Guo et al., 2021, Chen et al., 2020a]. **4)** As the tables show, **VMDLS** always outperforms **VMDLS$_b$**. This is also true for individual OOD sets, not just the averages. Thus, even if sometimes the improvement from **VMDLS$_b$** to **VMDLS** is small, we always recommend the latter.

### 4.1 ABLATION STUDY

**Losses.** We trained the model on MNIST several times each time using a different loss: (a) $\mathcal{L}_t$; (b) $\mathcal{L}_t + \mathcal{L}_d$ with $M_d = 32$; (c) $\mathcal{L}_{\mathrm{KL}} + \mathcal{L}_t$; (d) $\mathcal{L}_{\mathrm{KL}} + \mathcal{L}_t + \mathcal{L}_d$, $M_d = 1$; (e) $\mathcal{L}_{\mathrm{KL}} + \mathcal{L}_t + \mathcal{L}_d$, $M_d = 32$. The last two options are the proposed loss (with different $M_d$ values). In this ablation study, we chose $d = 2$ to let us visualize $\boldsymbol{\mu}$ directly, *without having to resort to a t-SNE visualization which might distort the structure*. Figure 4 depicts the values of $(\boldsymbol{\mu}(\boldsymbol{x}_i))_{i=1}^N$. Using only $\mathcal{L}_t$ (Figure 4a) or using $\mathcal{L}_t + \mathcal{L}_d$ without $\mathcal{L}_{\mathrm{KL}}$

Table 3: Macro-average F1-scores of 11 classes (10 known; 1 unknown) score on MNIST as the known classes and 3 different OOD sets

| Method | Omniglot | MNIST-noise | Noise |
|---|---|---|---|
| SoftMax | 0.595 | 0.801 | 0.829 |
| OpenMax | 0.780 | 0.816 | 0.826 |
| CROSR | 0.793 | 0.827 | 0.826 |
| CGDL | 0.850 | 0.887 | 0.859 |
| CPGM-AAE | 0.872 | 0.865 | 0.872 |
| CVAECapOSR | 0.971 | 0.982 | 0.982 |
| VMDLS$_b$ (Ours) | 0.969 | 0.961 | 0.963 |
| VMDLS (Ours) | **0.974** | **0.984** | **0.984** |

Table 4: Marco-average F1 scores with several losses. Known classes: MNIST. Unknown: Omniglot. $M_t = 0.1$ unless stated otherwise.

| Losses | F1-score |
|---|---|
| $\mathcal{L}_{\mathrm{KL}} + \mathcal{L}_t + \mathcal{L}_d$ ($M_d = 32$) | **0.969** |
| $\mathcal{L}_{\mathrm{KL}} + \mathcal{L}_t + \mathcal{L}_d$ ($M_d = 1$) | 0.941 |
| $\mathcal{L}_{\mathrm{KL}} + \mathcal{L}_t$ | 0.860 |
| $\mathcal{L}_t + \mathcal{L}_d$ ($M_d = 32$) | 0.935 |
| $\mathcal{L}_t$ | 0.858 |
| $\mathcal{L}_t$ ($M_t = 1$) | 0.829 |

(Figure 4b) leads to small distances between the classes whose clusters also differ from each other drastically in shape and size. Using $\mathcal{L}_{\mathrm{KL}} + \mathcal{L}_t$ (Figure 4c) leads to better results, but the clusters are far from being isotropic and, worse, are still close to each other in the vicinity of the origin. Using the proposed loss with either $M_d = 1$ (Figure 4d) or, even more so, $M_d = 32$ (Figure 4e), leads to nearly-isotropic clusters of similar sizes that are also much further away from each other. Table 4 summarizes the quantitative results obtained with **VMDLS$_b$** using the different losses on the MNIST-as-in-distribution benchmark, again with $d = 2$. We avoided here using **VMDLS** (which got slightly-better results – which we omit – than **VMDLS$_b$**) to focus on the loss aspect. Table 4 also includes the (inferior) result of using $\mathcal{L}_t$ with $M_t = 1$ (higher $M_t$ values were even worse).

**Feature Ensemble.** To study the effect of the feature ensemble, we compared **VMDLS** with different versions of itself where each version uses a different subset of the features, and evaluated the F1 performance of the different models in the OOD task where CIFAR-10 served as the in-distribution data while ImageNet-resize acted as the OOD data. As Table 5 shows, while the results were reasonably-good even when merely using $\boldsymbol{\mu}$, the added features helped and the best results were achieved when using the entire ensemble.

Table 5: The effect of the feature ensemble. In-distribution data: CIFAR-10. OOD data: ImageNet-resize.

| Features | $(\boldsymbol{t}^1, \boldsymbol{t}^2, \boldsymbol{t}^3, \boldsymbol{g}, \boldsymbol{\mu})$ | $(\boldsymbol{t}^2, \boldsymbol{t}^3, \boldsymbol{g}, \boldsymbol{\mu})$ | $(\boldsymbol{t}^3, \boldsymbol{g}, \boldsymbol{\mu})$ | $(\boldsymbol{g}, \boldsymbol{\mu})$ | $\boldsymbol{\mu}$ |
|---|---|---|---|---|---|
| *F1-score* | **0.927** | 0.925 | 0.919 | 0.918 | 0.906 |

## 5 CONCLUSION

We proposed an end-to-end SOTA method, for OOD detection, which requires no OOD data for learning/tuning. Its success lies in the good inter-class separation in the latent space, which also translates to a large empty area between the known classes where the OOD points end up. Our method has 2 main limitations. While some methods use a pre-trained DNN as is, ours requires either full DNN training or fine-tuning a pretrained DNN. However, besides

the fact that this is also the case with most recent methods, our SOTA results justify the training effort. The second limitation is that our training is about 4x slower than the training of a simple SoftMax-based classifier with the same backbone. However, this limitation can be mitigated considerably by using a pre-trained backbone (as we did in a few of the experiments in § 4) followed by fine tuning.

An interesting future direction which we have not explored in this work is how to incorporate uncertainty decomposition within the proposed method in order to improve the OOD detection, as was done in Bayesian neural networks and/or ensemble approaches [Vadera et al., 2020b, Malinin et al., 2019, Vadera et al., 2020a, Depeweg et al., 2018].

## Acknowledgements

This work was supported by the Lynn and William Frankel Center at BGU CS, by the Israeli Council for Higher Education via the BGU Data Science Research Center, and by Israel Science Foundation Personal Grant #360/21. O.D. was also funded by the Jabotinsky Scholarship from Israel's Ministry of Technology and Science, and by BGU's Hi-Tech Scholarship.

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
