# OpenReview forum: "Variational- and Metric-based Deep Latent Space for Out-of-Distribution Detection"
_auai.org/UAI/2022/Conference — UAI 2022 Poster_

### Official Review · Reviewer_99cU · 2022-04-12

**Q2(1) Originality/Novelty:** 3
**Q2(2) Significance/Impact:** 3
**Q2(3) Correctness/Technical Quality:** 3
**Q2(6) Clarity Of Writing:** 3
**Q6 Overall Score:** 6
**Q8 Confidence In Your Score:** 4

**Q1 Summary And Contributions:**

This paper presents an OOD detection approach, by training a new model based on the backbone of a classifier to project data to the latent space, and train the model using a combination of KL loss, a trip loss and a new distancing loss. The paper demonstrates that this results in feature ensemble for individual classes to have close to isotropic Gaussian distribution, which are far away from each other. Additionally, the paper shows that OOD feature ensemble lies in the middle of per class maps.

**Q2 Assessment Of The Paper:**

More detailed information regarding each of these aspects is given below:

**Q2(4) Quality Of Experiments (Optional):**

3: Good: The experimental evaluation is adequate, and the results convincingly support the main claims.

**Q2(5) Reproducibility:**

4: Excellent: Key resources (e.g., proofs, code, data) are available and key details (e.g., proof sketches, experimental setup) are comprehensively described for competent researchers to confidently and easily reproduce the main results.

**Q3 Main Strengths:**

The main strength of the paper is that the approach is relatively simple and builds upon existing work on training using triplet losses and modeling per class feature ensemble using Gaussian distributions. The authors have also done a lot of experiments comparing it with existing approaches on OOD detection across a range of datasets.

**Q4 Main Weakness:**

It seems that the backbones for different baselines across experiments aren't consistent with what's being used in the proposed method. Additionally, the authors leave out all discussion around uncertainty decomposition and its applications to OOD detection using Bayesian neural networks/ensemble approaches.

**Q5 Detailed Comments To The Authors:**

- It's not immediately clear how the KL loss is helping ensure that per class feature ensembles would form an isotropic Gaussian. Could you elaborate on this and provide an intution?

- It seems like VMDLS does require you to choose a hyperparameter (\delta) for final detection, however the abstract claims that the approach requires no hyperparam tuning. Could you please elaborate on this?

- For the F1 score computation in OSR setting, how does the model make class predictions? By looking at max log likelihood of belonging to either of the C isotropic Gaussians of the feature ensemble?

- How do we come up with M_t and M_d values? How would you advise users to about selecting them?

- What's the intuition behind choose t^1, t^2, and t^3? In practical sense, how deep should we go while choosing feature maps, and it would seem that the dimensionality of the vector t_i would be very high. Can we do something more efficient here?

- The paper should include discussion around uncertainty decomposition and its applications to OOD detection using Bayesian neural networks/ensemble approaches, as these uncertainty scores can be better at OOD detection than some of the baselines (see refs [1-4]).


References

[1] Vadera, Meet P., Adam D. Cobb, Brian Jalaian, and Benjamin M. Marlin. "Ursabench: Comprehensive benchmarking of approximate bayesian inference methods for deep neural networks." arXiv preprint arXiv:2007.04466 (2020).

[2] Malinin, Andrey, Bruno Mlodozeniec, and Mark Gales. "Ensemble distribution distillation." ICLR (2019).

[3] Vadera, Meet, Brian Jalaian, and Benjamin Marlin. "Generalized bayesian posterior expectation distillation for deep neural networks." UAI (2020).

[4] Depeweg, S., Hernandez-Lobato, J., Doshi-Velez, F. & Udluft, S. Decomposition of Uncertainty in Bayesian Deep Learning for Efficient and Risk-sensitive Learning. ICML (2018).


**Q7 Justification For Your Score:**

Overall, I am happy with the paper and inclined towards an accept. I'd also like to note that the authors have included their implementation, which is admirable.

I am open to updating my score based on authors response to my questions and discussions with the other reviewers.

**Q9 Complying With Reviewing Instructions:**

1: Yes.

---

### Official Review · Reviewer_26Yi · 2022-04-12

**Q2(1) Originality/Novelty:** 2
**Q2(2) Significance/Impact:** 3
**Q2(3) Correctness/Technical Quality:** 4
**Q2(6) Clarity Of Writing:** 4
**Q6 Overall Score:** 8
**Q8 Confidence In Your Score:** 4

**Q1 Summary And Contributions:**

The authors propose an OOD detection systems that combine ideas taken from VAE and Contrastive Learning to better separate classes away from each other. During inference, the authors use a weighted log-likelihood to decide whether the data belongs to one out of C classes or if it is an OOD point. To show the effectiveness of their approach, the authors experiment the model on popular image benchmarks and compare it with several SoTA methods proving that their model yields SOTA results.


**Q2 Assessment Of The Paper:**

More detailed information regarding each of these aspects is given below:

**Q2(4) Quality Of Experiments (Optional):**

3: Good: The experimental evaluation is adequate, and the results convincingly support the main claims.

**Q2(5) Reproducibility:**

4: Excellent: Key resources (e.g., proofs, code, data) are available and key details (e.g., proof sketches, experimental setup) are comprehensively described for competent researchers to confidently and easily reproduce the main results.

**Q3 Main Strengths:**

The main strengths of the paper is (1) the use of a new metric-loss term that helps to better separate classes away from each other and (2) the use of the weighted log-likelihood to improve the decision rule in the testing phase.


**Q4 Main Weakness:**

The use of a user-defined threshold in the decision rule (to decide whether a point belongs to one of C classes or if it is an OOD points).


**Q5 Detailed Comments To The Authors:**

In the SupMat, Datasets section, I suggest to insert a brief description of the used datasets: e.g., MNIST is a dataset of handwritten digits in which each instance consists of a 28x28 gray-scale image representing a digit in the interval {0, …, 9}.
As for CIFAR and MNIST, the dataset consists of training set and test set, in the experiments did you use only the data provided in the training set or also those of the test set? Specifically, e.g., for MNIST the training set consists of 60K examples and the test set of 10K: In the learning phase, did you only use the training set, or did you combine the two sets (70K)? You could also include these details in the description.

Typos:
Paper
	•	Page 5, Eq. 6: $\sum_{j=1}^{d} -\log \frac{s}{\sigma_j (g(x_i))} - \frac{\sigma_j(g(x_i)) + (\mu_j(g(x_i)) - m_j(y_i))^2}{2s^2} + \frac{1}{2}$ should be $\sum_{j=1}^{d} -\log \frac{s}{\sigma_j (g(x_i))} - \frac{\sigma_j(g(x_i))^2 + (\mu_j(g(x_i)) - m_j(y_i))^2}{2s^2} + \frac{1}{2}$
	•	Page 6: “For some intuition, suppose that that in-distribution points are cars and the OOD points are animals.” → “For some intuition, suppose that in-distribution points are cars and the OOD points are animals.”

SupMat
	•	Page 2: Eq. 13: $...= \sum_{j=1}^{d} \log \bigg( \frac{s}{\sigma_j (g(x_i))} \bigg) + \frac{\sigma_j(g(x_i))^2 + (\mu_j(g(x_i) - m_j(y_i))^2}{2s^2} - \frac{1}{2}$ → $...= \sum_{j=1}^{d} \log \bigg( \frac{s}{\sigma_j (g(x_i))} \bigg) + \frac{\sigma_j(g(x_i))^2 + (\mu_j(g(x_i)) - m_j(y_i))^2}{2s^2} - \frac{1}{2}$
	•	Page 2: Eq. 14: $...= \sum_{j=1}^{d} \bigg[ -\log \bigg( \frac{s}{\sigma_j (g(x_i))} \bigg) - \frac{\sigma_j(g(x_i)) + (\mu_j(g(x_i)) - m_j(y_i))^2}{2s^2} + 0.5 \bigg]$ should be $...= \sum_{j=1}^{d} \bigg[ -\log \bigg( \frac{s}{\sigma_j (g(x_i))} \bigg) - \frac{\sigma_j(g(x_i))^2 + (\mu_j(g(x_i)) - m_j(y_i))^2}{2s^2} + 0.5 \bigg]$
	•	Page 5: “Thus the choice of low-level features is the the optimal for our use case.” —> “Thus the choice of low-level features is the optimal for our use case.”


**Q7 Justification For Your Score:**

The paper is well-organized and well-written. The idea is very interesting, especially the aid of a new metric loss that better separates the classes and at the same time creates, between the known classes, an empty space wherein the most of the OOD data end ups. I think that the paper should be accept.


**Q9 Complying With Reviewing Instructions:**

1: Yes.

---

### Official Review · Reviewer_3dRq · 2022-04-13

**Q2(1) Originality/Novelty:** 2
**Q2(2) Significance/Impact:** 1
**Q2(3) Correctness/Technical Quality:** 2
**Q2(6) Clarity Of Writing:** 2
**Q6 Overall Score:** 4
**Q8 Confidence In Your Score:** 3

**Q1 Summary And Contributions:**

This paper introduces a new set of loss terms to facilitate OoD detection. The basic idea is to train a classifier with in-domain samples while keeping them well separable in the latent space as isotropic Gaussians.

**Q2 Assessment Of The Paper:**

More detailed information regarding each of these aspects is given below:

**Q2(4) Quality Of Experiments (Optional):**

3: Good: The experimental evaluation is adequate, and the results convincingly support the main claims.

**Q2(5) Reproducibility:**

2: Fair: Key resources (e.g., proofs, code, data) are unavailable but key details (e.g., proof sketches, experimental setup) are sufficiently well-described for an expert to confidently reproduce the main results.

**Q3 Main Strengths:**

Although the proposed solution is incremental improvement based on SOTA, the authors did a good job in motivating and explaining most of those components in terms of its contribution. Especially the section 4.1. provides a fairly good ablation study. The experiments are presented very clearly and also serve to support the major claim.

**Q4 Main Weakness:**

For technical issues please refer to Q5.
There are just too many typos that have to be corrected. To name a few "Gaussian-based approached" -> "Gaussian based approaches" on page 2. "nearly-istoropic" -> "nearly-isotropic". "the a likelihood-based" on page 3 etc. etc.

**Q5 Detailed Comments To The Authors:**

The L_d loss is sufficiently motivated but I wonder if one could achieve the same effect by simply defining weight for the term $f(g(x^a))-f(g(x^n))$. As the authors already pointed out, we only need one of $M_t$ and $M_d$, at least empirically.
The only doubt I have on the proposed method is with the KL loss. Specifically, the mean of the distribution $m(c)$ conditioned on the class seems to be a smoothing regularization preventing the latent representations change too much in the course of the training. The idea itself makes sense but I fail to understand how this term improves the separability of the Gaussians (such as Fig 4 b)). I would be very interested in some more explanation in this regard.


**Q7 Justification For Your Score:**

My major concern is that the contribution is very much incremental (although probably practical) and may not result in a high impact in the research field.

**Q9 Complying With Reviewing Instructions:**

1: Yes.

---

### Official Review · Reviewer_PE6m · 2022-04-15

**Q2(1) Originality/Novelty:** 2
**Q2(2) Significance/Impact:** 2
**Q2(3) Correctness/Technical Quality:** 2
**Q2(6) Clarity Of Writing:** 3
**Q6 Overall Score:** 5
**Q8 Confidence In Your Score:** 4

**Q1 Summary And Contributions:**

This paper proposes learning a metric space jointly with a classifier network for out-of-distribution detection without using any OOD sample for training or fine-tuning. They propose using a triplet loss, a KL variational loss and distance loss, which help to shape the metric space and have classes be further apart and more isotropic Guassian-like. They show competitive performance on a number of OOD benchmarks versus the SOTA methods.

**Q2 Assessment Of The Paper:**

More detailed information regarding each of these aspects is given below:

**Q2(4) Quality Of Experiments (Optional):**

2: Fair: The experimental evaluation is weak: important baselines are missing, or the results do not adequately support the main claims.

**Q2(5) Reproducibility:**

3: Good: Key resources (e.g., proofs, code, data) are available and key details (e.g., proofs, experimental setup) are sufficiently well-described for competent researchers to confidently reproduce the main results.

**Q3 Main Strengths:**

The paper addresses an important open problem.

It presents some interesting ideas for regularizing the learned metric space and shows nice empirical results and visualizations to support how their mathematical formulation alters the metric space.

The results of the algorithm are better than the SOTA.

**Q4 Main Weakness:**

- "increasing $M_t$ in the triplet loss would result in positive examples ending up too far from the anchor". This is not always true. It can also happen that the network keeps the positives where they are and simply pushes the negative examples out to satisfy the increased$M_t$. Furthermore, if the distance loss affects a change in the anchors, it can increase the distance between the anchor and positive as well. So, the authors line of reasoning is not entirely correct and they should remove statements not guaranteed to hold true.

- It is also not clear as to why the above losses should ensure a large empty space between all the gaussians? There is nothing restricting  the learned mean of any class from not all having the value of zero.

In Table 2, the best performing methods (CVAECapOSR and ODIN) don't use the same baseline network as the proposed method. Hence it is hard to compare them fairly.
In Table 3, the proposed method is only marginally better than the SOTA (CVAECapOSR).


**Q5 Detailed Comments To The Authors:**

- "However, the problem with that Gaussian-based approached" --> "However, the problem with that Gaussian-based approach"
- section 3: "achieved, the a likelihood-based" --> "achieved, a likelihood-based"
- In equation 1, the Hadamard product should only be between $\sigma$ and $\eta$
- "within out method" --> "within our method"
- Is m(c) the same as m(y_i) used in equation 5? If so, why the notational change from $y_i$ to $c$?




**Q7 Justification For Your Score:**

Overall, the paper presents some novel ideas for how to learn a metric space for OOD detection. The ideas are mostly borrowed from several previous works and combined together in a new way for solving the task at hand. The intuitive explanation of the method is hand-wavy and not very convincing. However, the results are competitive with the SOTA and slightly better at times. Hence, I am leaning positive, but would not be opposed to rejection.

**Q9 Complying With Reviewing Instructions:**

1: Yes.

---

### Decision · Program_Chairs · 2022-05-15

**Decision:**

Accept (Poster)

**Comment:**

Meta Review: This paper proposes some modifications to [Lee et al., 2018b]'s methodology of Gaussian clustering of the latent space for OOD detection.  The reviewers were a bit split over the decision, with most finding the results compelling (if not impressive) but the proposed modifications incremental.  I agree yet ultimately find that the paper's practical merits warrant publication.